# A Comparative Analysis of Functional Status and Mobility in Stroke Patients with and without Aphasia

**DOI:** 10.3390/jcm11123478

**Published:** 2022-06-16

**Authors:** Zbigniew Guzek, Wioletta Dziubek, Małgorzata Stefańska, Joanna Kowalska

**Affiliations:** 1Department of Neurological Rehabilitation, University Hospital in Zielona Góra, 65-046 Zielona Góra, Poland; guzek@o2.pl; 2Faculty of Physiotherapy, University of Health and Sport Sciences, 51-612 Wroclaw, Poland; malgorzata.stefanska@awf.wroc.pl (M.S.); joanna.kowalska@awf.wroc.pl (J.K.)

**Keywords:** stroke, aphasia, functional outcome, balance, trunk control, rehabilitation, effectiveness of rehabilitation

## Abstract

All researchers agree that aphasia is a serious consequence of a stroke, but they also report contradictory data regarding the functional outcome. The aim of this study was, therefore, to assess the functional outcomes of stroke patients with and without aphasia, who were undertaking a regular rehabilitation programme. Materials and Methods: The study group consisted of 116 post-stroke patients, including 54 patients without aphasia (G1) and 62 patients with aphasia (G2). The following tests were used before (T1) and after (T2) rehabilitation measurement points: Barthel Index (BI), Sitting Assessment Scale (SAS), Berg Balance Scale (BBS), Trunk Control Test (TCT), Test Up & Go (TUG) and the Timed Walk Test (TWT). Results: The group of post-stroke patients with aphasia had a significantly longer time since a stroke on admission, a significantly longer length of stay in the ward and significantly worse SAS and TCT scores at T2, compared to patients without aphasia. Both groups achieved significant improvement in all studied parameters (SAS, TCT, BI, BBS, TUG and TWT). Aphasia was a predictor of functional status in the stroke patients group, but only at the time of admission to the ward. Conclusions: Patients with and without aphasia have an equal likelihood of improving their functional status and returning to independence. Aphasia should not be an absolute factor that excludes stroke patients from research studies on their functional status.

## 1. Introduction

Despite the decreasing incidence of mortality, the socioeconomic burden of stroke has increased and is likely to remain high [1]. The prevalence of people living with the effects of stroke has increased because of the growing and ageing population. The increasing number of stroke survivors creates a greater demand for rehabilitation services [2]. Rehabilitation is a necessary non-pharmacological treatment that is used after a stroke to improve functional status and return to independence and so it is important to identify and consider any factors that affect rehabilitation and examine existing factors in more detail. 

According to current studies, there are many factors that determine the effectiveness of rehabilitation in stroke patients. These include age, type of stroke, presence of cognitive impairments and post-stroke depression, acceptance of the disease, level of self-sufficiency, functional status at admission to a rehabilitation ward, motor function and trunk movements [3,4,5,6,7,8,9]. The presence of aphasia is also relevant [9,10,11].

Aphasia is a common neuropsychological deficit in patients who have had a stroke [12]. It is a communication disability disorder caused by brain damage that affects verbal communication and the production or understanding of speech, reading and writing. Aphasia is damage to the left cerebral hemisphere, is usually secondary to a stroke and affects people differently depending on the injured brain area [12,13]. It occurs in 21–38% of stroke patients [14,15,16].

According to some authors, aphasia can affect post-stroke functional recovery and clinical outcome [17], and patients with post-stroke aphasia have worse upper extremity motor dysfunction [18]. Others suggest that aphasia is an obstacle to rehabilitation and is a predictor of motor, functional and social outcomes [12,19]. Some researchers argue that the presence of aphasia has a small but clinically irrelevant effect on functional improvement [20]. Stroke patients with aphasia are also more likely to have depression, memory disorders and longer stays necessary for rehabilitation [12,21,22]. Aphasia may be a risk factor in the development of post-stroke depression [23] and it can be a prognostic factor for cognitive status [10]. 

Aphasia may negatively affect the rehabilitation process via several mechanisms; for example: it can prevent a patient from understanding therapeutic instructions; if the patient cannot follow the rehabilitation therapist, they may begin to spontaneously exercise the affected area; motor apraxia often accompanies aphasia and interferes with motor activities, thus impairing functional recovery [12,21]. It is worth noting that the reported findings regarding the functional status of stroke patients with aphasia also depend on the type of aphasia present [12,19,24]. 

All researchers agree that aphasia is a serious consequence of stroke; however, they report contradictory data regarding the functional status and recovery of patients with and without aphasia after a stroke. The role of aphasia in predicting rehabilitation outcomes after a stroke has not been clearly defined. This may be because stroke patients with aphasia are routinely excluded from participation in some areas of stroke research [3,5,24]. Patients who are excluded because of communication difficulties, either due to aphasia or language barriers, are an understudied subset of the stroke population [2,3,5].

The aim of this study was, therefore, to assess functional status and its improvement in a group of post-stroke patients with and without aphasia, undergoing systematic rehabilitation and to identify factors affecting the functional status of post-stroke patients at the time of admission and at discharge.

Understanding the role of aphasia in the rehabilitation process is important for care, stay at the ward, treatment planning and improving the functional status of stroke patients. Stroke patients with aphasia are an important and large group of patients, significant not only from the researcher’s point of view, but also that of the practicing therapist.

The results of the study may have utilitarian implications, leading to a change in therapy regimens at physiotherapy centers treating people after stroke. 

## 2. Materials and Methods

### 2.1. Studied Group

This study was conducted at the Neurological Rehabilitation Unit of the Department of Rehabilitation at the University Hospital in Zielona Góra, Poland, with the consent of the head of the department and under the ethical and legal supervision of the Bioethics Committee of the University of Health and Sport Sciences in Wroclaw, Poland (reference no. 16/2021). The study was conducted in accordance with the Helsinki Declaration.

The study group consisted of post-stroke patients who were consecutively admitted to a rehabilitation ward in the period from January 2019 to February 2022 and who satisfied the following inclusion criteria: written informed consent to participate in the study, first stroke incident, patients admitted from the hospital’s neurological unit, patients who were able to understand speech and commands well (including patients with, e.g., expressive aphasia, Broca’s aphasia), no severe dementia and left hemispheric stroke. Exclusion criteria were also established to include: the occurrence of aphasia with speech comprehension problems, refusal to participate at any stage of the study and the presence of serious mental disorders (e.g., consciousness disorders or mental disorders) according to the medical records, or at the time of the study.

The analysed data was sourced from 116 patients with a mean age of 68.3 (±11.3) years; 50 women and 66 men. The majority of patients in the studied group had suffered ischemic stroke (85%). 

Respondents were divided into two groups according to the occurrence of aphasia (based on the assessment of a speech therapist and available medical records):
G1:54 patients without aphasia;G2:62 patients with aphasia.

Detailed patient characteristics are presented in Table 1 and Table 2. 

### 2.2. Measurement Tools

Post-stroke patients were assessed using the following measurement scales: the Barthel Index (BI), the Sitting Assessment Scale (SAS), the Berg Balance Scale (BBS), the Trunk Control Test (TCT), the Test Up & Go (TUG) and the Timed Walk Test (TWT). Sociodemographic and clinical data were collected based on the interviews and previous medical records.

The BI is an activities of daily living (ADL) scale, which assesses the patient’s performance in ten basic everyday activities, such as eating, moving about, dressing and using the toilet. The maximum a patient can obtain is 100 points. Scores from 0–20 indicate a severe condition, 21–85 a medium-heavy condition and between 86–100 the patient’s condition is described as mild [25].

The SAS is a method of assessing sitting ability by analysing head, trunk and foot control, as well as arm and hand function. The patient’s sitting balance is scored as: 4—able to perform the above tests without any physical assistance; 3—able to maintain a static position without difficulty, however requiring assistance, especially in righting from the hemiplegic side; 2—able to maintain a static position but requiring assistance in all righting tasks; and 1—unable to maintain a static position [26,27].

The BBS consists of 14 tasks, which are used to evaluate a patient’s static and dynamic balance. These tasks include: sitting unsupported, standing unsupported, standing with eyes closed, standing with feet together, standing on one foot, turning to look behind, retrieving an object from the floor, tandem standing, reaching forward with an outstretched arm, sitting to standing, standing to sitting, transferring and turning 360° and then stepping on a stool. A score from 0 to 4 is given for each task. The maximum score is 56 in total. The higher the total score, the better the balance [28,29].

The TCT assesses four aspects of trunk movement: rolling on a patient’s weak and strong sides, sitting up from lying down and sitting in a balanced position on the edge of the bed, with feet off the ground. Patients are given the following scores: 0—unable to perform movement without assistance; 12—able to perform movement, but in an abnormal way; and 25—able to complete movement normally. Maximum score equals 100 in total [19,30]. 

The TUG test is used to assess functional ability and fall risk. The TUG test requires patients to stand up from a chair, walk 3 m, turn around, walk back to the chair and sit down. The timing of the test begins at the word “go”, and ends when the participant is seated. The patient may use supporting equipment during the test [31,32].

The TWT (5 m) assesses the gait speed of patients over a distance of 5 m. They have to walk 5 m as quickly as possible. It is a crucial prognostic factor for older adults, and it is recommended for the assessment of longitudinal change in walking ability after a stroke. A higher gait speed indicates better walking ability [7,33].

The above tests were performed at two measurement points: T1, on the first day of a patient’s admission to the rehabilitation ward (initial assessment); T2, the final assessment, on the last day of their stay on the ward. 

Both groups of patients took part in regular rehabilitation. The rehabilitation programme was carried out in accordance with a doctor’s instructions. It was performed to a specified frequency and duration: from Monday to Friday for about 150 min per day and 90 min on a Saturday. This programme was dependent on the functional status of the patient and included individual exercises with a physiotherapist (120 min) and activities with the aim of learning and improving gait (30 min, e.g., walking on a flat and uneven surface, walking on a special learning track and learning to walk up and down stairs). Every patient had an occupational therapist and patients with aphasia also had speech therapy three times a week for 45 min. Prior to admission to the rehabilitation ward, all patients were subjected to early post-stroke rehabilitation on the hospital ward.

### 2.3. Data Analysis

The following descriptive statistics were calculated for the analysis: mean, standard deviation, median and interquartile range. A Shapiro Wilk test did not confirm the normality of distribution for most variables measured on the quantitative scale. The significance of differences between groups was tested using the Mann–Whitney U test, Chi-square test and Fisher’s exact test.

The Kruskal–Wallis ANOVA test was applied in order to assess the significance of differences between the results obtained in the two groups regarding the measurement number. When the analysis of variance revealed statistical significance, a multiple comparison of mean ranks test was used as a post hoc test. Multivariate regression analysis was performed to identify associations between the BI test results and other selected parameters. A corrected Cohen’s d test was used to determine the quantity of the effect of differences between examined groups. The effect size of interaction was calculated by Eta squared (η2) and then transformed to Cohen’s d value [34]. The values of the Cohen’s d test ≥ 0.8 demonstrated the high strength of the observed effect. Multiple regression effect size references were computed using the Cohen’s f-square test [35].

All calculations were made using the Statistica 13.1 and statistical calculators at: http://www.psychometrica.de/effect_size (accessed on 21 March 2022) and https://www.analyticscalculators.com/calculator.aspx?id=5 (accessed on 21 March 2022).

## 3. Results

The structure of the two groups was similar (Table 1); however, significant differences between groups were found in the time elapsed since stroke and the length of stay in the rehabilitation ward. The mean time since stroke and the mean length of stay in the rehabilitation ward were both longer for the G2 group than for the G1 group (Table 2). 

A comparative analysis of the studied parameters at admission (T1) showed statistically worse SAS (trunk, lower limb, upper limb) and TCT scores in the group of patients with aphasia (G2). At T2, however, the two groups significantly differed only in SAS (lower limb, upper limb) (Table 3 and Table 4).

There was a statistically significant improvement between the initial and final tests in SAS (trunk, arm control, hand function), BI and BBS in both groups. A statistically significant improvement was also achieved in TCT in the aphasia group (Table 3 and Table 4).

A qualitative data analysis was performed because not all patients were able to perform the TUG and TWT tests in both the initial and final study. It showed that in the initial study (T1), more than 68% (*n* = 37) of G1 patients and 79% (*n* = 49) of G2 patients did not perform the TUG test due to their functional status. In the final examination (T2), only 24% and 35% of patients in G1 and G2, respectively, failed the test. This change was statistically significant. The TWT test had similar results (Table 5).

The linear regression analysis revealed that aphasia had a significant effect on the BI scores at the beginning of a patient’s rehabilitation (T1) and no effect on the BI scores at T2. Consistent with the model at T1, patients with aphasia scored 15 points lower than patients without aphasia (Table 6).

Multivariate regression analysis showed that age, marital status and the presence of aphasia were the socio-demographic factors with the greatest impact on BI scores at the commencement of rehabilitation (T1). Age and gender continued to have a significant effect on the BI scores at the time of a patient’s discharge from the ward (T2) (Table 7).

The regression analysis revealed that, of the parameters assessing the functional status of stroke patients, it was the BBS score that had the greatest effect on BI at T1 and at T2, and the TCT score at T2 (Table 8).

## 4. Discussion

The primary objective of rehabilitating patients after a stroke is the restoration of independence, with aphasia being one of the many factors involved in the functional outcome. There were no significant differences regarding demographic variables between patients with aphasia and those without aphasia at baseline. Similar results were achieved by Hilari [36]. 

Several statistically significant differences were observed in the clinical and functional data. At the time of admission, patients with aphasia were characterised by a significantly longer time elapsed since stroke and significantly longer length of stay in the rehabilitation ward compared to patients without aphasia. These results are consistent with reports by other authors. Paolucci et al., reported the mean length of stay for rehabilitated patients with aphasia as 14 days longer than those without aphasia [37]. In our study, this difference was on average 15 days. Gialanella and Prometti demonstrated that aphasia is an important predictor of length of stay for rehabilitation [21]. Different results were reported by García-Rudolph et al., who observed no significant differences in length of stay between patients with and without aphasia [11].

On admission to the ward, the group of patients with aphasia had worse scores for trunk control (TCT) and the results in the SAS: trunk, arm control and hand function, compared to patients without aphasia. Similar results for the TCT scale were obtained by Gialanella et al. [12,19] and Xu et al., who emphasised that stroke patients with aphasia have worse upper extremity motor function compared to those without aphasia [18].

It is worth noting that despite the worse results, the ability of patients to perform basic daily activities (as measured by the BI) was comparable at baseline and did not differentiate between the two study groups. The same results were obtained when comparing balance (BBS). In the final study, significant differences were found between the groups only in the SAS arm control and SAS hand function scores. Different results were reported by Seo et al., where the BI scores were significantly lower in aphasic patients than in non-aphasic controls [10]. Significantly worse results were found for the group of patients with aphasia, concerning everyday activities (measured with the Functional Independence Measurement), in studies by other authors, both at baseline and at the end of the study [12,19]. Furthermore, Hilari showed that patients with aphasia performed significantly worse than a comparable group of people without aphasia: looking at the extended ADLs (measured with the Frenchay Activities Index—FAI) that were particularly affected for patients with aphasia, she found that this was not related to physical activities such as performing housework or going for a walk, but rather social, leisure activities and work, such as shopping, hobbies and travelling for pleasure; that is, activities which required communication [36]. These differences may, however, be attributable to the study’s adopted inclusion criteria. The patients with aphasia who were included in the study were those who did not have problems with the comprehension of speech, which probably affected the results.

Comparing the results of the initial and final examinations suggests that both study groups achieved significant improvements in almost all parameters (SAS, TCT, BI, BBS). The improvements in functional status, trunk control, mobility, walking ability and balance, and, above all, the better performance of daily living activities, testify to the effectiveness of rehabilitation, in stroke patients with aphasia as well as those without. Many authors have reported an improvement in all parameters after the rehabilitation period for stroke patients, including those with aphasia [7,8,12,17,37,38,39].

The effectiveness of rehabilitation and improvement in mobility and walking ability in both groups were also confirmed by the TUG and TWT test results. Following the rehabilitation, at the time of discharge, 28 and 34 patients, or 52% and 55% of the G1 (without aphasia) and G2 (with aphasia) groups, respectively, were walking independently without orthopaedic aids. In the initial study, this was only 16% of patients in both groups. Harvey reported, citing other researchers, that 70% to 80% of chronic stroke survivors have the ability to walk, but only 30% to 50% return to community ambulation [9]. Such efficacy in improving mobility and gait in both groups is associated with significant improvements in SAS, TCT and BBS scores. The results obtained are consistent with reports by other authors. Duarte et al., reported that a good sitting balance at onset predicts a return to independent walking, while a score of ≤50 on the TCT 14 days after stroke predicts that walking is unlikely at six-month follow-up [40]. Awad et al., noted that early sitting balance can predict a later return to walking and only gains in dynamic (walking) balance are associated with improvements in long-distance ambulation [41]. The strongest predictor of independent walking, however, is balance [9,23].

The regression analysis carried out on the entire study group of stroke patients did not confirm the significance of factors such as the length of time since a stroke, or length of stay, in the functional status of the subjects, especially at discharge. Studies by authors who reported a relationship between the functional outcome and length of stay [38] and time since stroke [34] have thus not been confirmed. It is also worth noting that the presence of aphasia was a predictor of functional status, but only at the time of admission to the ward. Such a significant relationship was not observed at the time of discharge, which is also contrary to studies by many authors [11,12], but confirms other reports [20,23].

Among the other factors investigated, age, marital status and BBS score were predictors of initial functional status. Age, gender, BBS score and TCT score were found to be significant predictors of a patient’s final functional status. These results are not surprising; researchers often point out that a younger age does predict better outcomes [6,7] and the male gender has also been associated with better functional outcome [6,9]. The best functional status at the time of admission was reported in subjects who were single, similar to the study by Szczepańska-Gieracha et al. [42].

Researchers confirm that balance is a stronger predictor of walking outcome and activities of daily living than age or motor strength [23]. Harvey emphasises that the ability to regain walking can be predicted by balance ability at onset [9]. Better trunk control predicts better ADL at discharge [30]. In the present study, the BBS and TCT scores explained as much as 78% of the variance in the dependent variable, that is the functional status at discharge (BI at T2). This confirms that balance and trunk control are strong predictors of the final functional status of stroke patients.

The significant improvement in the examined parameters of both groups of patients indicates that they have an equal chance of improving their functional status and returning to independence. Lower effectiveness thus cannot be assumed in the group of patients with aphasia. The results of studies so far are highly divergent, however. Most authors emphasise that aphasia is a strong predictor of functional outcome [11,12,19], which has not been confirmed by this study. Such varying results are probably associated with the small number of stroke patients with aphasia, differences in study methodology and inclusion criteria (e.g., type of aphasia in the subjects). This is not an obstacle to effective rehabilitation, however, or a reason to exclude these patients from future studies.

### Limitations

There were some limitations to this study. First of all, it is a single-centre study, so the results should not be generalised. Patients with aphasia (mainly Broca’s aphasia; patients with no speech comprehension problems) were included in the study without taking into account other types of aphasia and aphasia severity. The size of the study groups would need to be increased in order to strengthen the conclusions and confirm the results of patients with and without aphasia, and to identify predictors of functional outcome among patients with aphasia. In the future, the presence of hemineglect and spasticity, for example, should be taken into account. It is impossible to take into account all factors affecting the functional outcome, and, therefore, in this study we analysed only those parameters that are typically tested at the time of a patient’s admission to a post-stroke rehabilitation ward and at the time of discharge. In most cases, the selection corresponded with the studies of other authors.

## 5. Conclusions

The examined group of post-stroke patients with aphasia had a significantly longer time since stroke on admission and a significantly longer length of stay on the ward, as well as significantly worse SAS and TCT scores on admission compared to patients without aphasia.

There were statistically significant improvements in functional status, mobility, walking ability and balance in both groups investigated (with and without aphasia).

In the group of stroke patients, aphasia was a predictor of functional status, but only at the time of admission.

Of the factors considered, age, gender, balance and trunk control were predictors of better functional status at discharge.

## Figures and Tables

**Table 1 jcm-11-03478-t001:** Characteristics of patients in all study groups and in subgroups (χ^2^ test).

Variable		All		G1		G2		*p*
		*n*	%	*n*	%	*n*	%	
Gender	Female	60	43.1	32	59.3	28	45.2	0.6315
	Male	56	56.9	22	40.7	34	54.8	
Education	Secondary and higher	72	62.1	32	59.3	40	64.5	0.5607
	Primary and vocational	44	37.9	22	40.7	22	35.5	
Marital status	Single (widow(er), unmarried)	64	55.2	28	51.9	36	58.1	0.5021
	Married	52	44.8	26	48.1	26	41.9	
Type of stroke	Ischemic	99	85.3	45	83.3	54	87.1	0.5675
	Haemorrhagic	17	14.7	9	16.7	8	12.9	

G1—patients without aphasia; G2—patients with aphasia

**Table 2 jcm-11-03478-t002:** Characteristics of patients in all study groups and in subgroups (the Mann–Whitney U test).

Parameters	All *n* = 116		G1; *n* = 54		G2; *n* = 62		*p*	Cohen’s d
	Median	IQR	Median	IQR	Median	IQR		
Age (years)	69.00	15.00	71.00	16.00	67.00	16.00	0.2560	0.21
Time since stroke (days)	15.00	9.00	13.50	8.00	15.00	11.00	0.0270 *	0.32
Length of stay (days)	63.00	58.00	46.00	51.00	76.00	56.00	0.0159 *	0.47

IQR—inter-quartile range; G1—patients without aphasia; G2—patients with aphasia * *p* < 0.05.

**Table 3 jcm-11-03478-t003:** Descriptive statistics of the studied parameters at T1 and T2.

Group	Parameters	T1				T2			
		Mean	SD	Median	IQR	Mean	SD	Median	IQR
G1	SAS head	3.80	0.5	4.00	0.00	3.94	0.2	4.00	0.00
	SAS trunk	3.39	0.8	4.00	1.00	3.80	0.7	4.00	0.00
	SAS arm control	2.98	1.1	3.00	2.00	3.56	0.9	4.00	0.00
	SAS hand function	2.69	1.2	3.00	2.00	3.46	0.9	4.00	1.00
	BI	35.93	25.1	37.50	50.00	76.02	31.7	92.50	40.00
	BBS	26.04	14.2	25.00	24.00	41.72	15.6	47.00	20.00
	TCT	87.33	20.7	100.00	26.00	95.20	14.9	100.00	0.00
G2	SAS head	3.65	0.6	4.00	1.00	3.92	0.3	4.00	0.00
	SAS trunk	2.84	1.0	3.00	2.00	3.68	0.6	4.00	1.00
	SAS arm control	2.05	1.1	2.00	2.00	2.95	1.1	3.00	2.00
	SAS hand function	1.92	1.0	2.00	2.00	2.85	1.2	3.00	2.00
	BI	20.56	21.6	10.00	30.00	63.23	32.4	70.00	55.00
	BBS	17.55	16.0	11.00	24.00	37.63	17.8	46.00	36.00
	TCT	67.35	28.6	67.50	52.00	89.35	19.1	100.00	13.00

G1—patients without aphasia; G2—patients with aphasia; BI—Barthel Index; SAS—Sitting Assessment Scale; BBS—Berg Balance Scale; TCT—Trunk Control Test; T1—initial assessment; T2—final assessment.

**Table 4 jcm-11-03478-t004:** Kruskal–Wallis analysis of variance—Post hoc test results.

	G1 Group T1 vs. T2	G2 Group T1 vs. T2	T1 G1 vs. G2	T2 G1 vs. G2	Cohen’s d
SAS head	1.0000	0.1238	1.0000	1.0000	0.82
SAS trunk	0.0262 *	<0.0001 *	0.0416 *	1.0000	1.66
SAS arm control	0.0428 *	0.0004 *	0.0006 *	0.0296 *	1.72
SAS hand function	0.0046 *	0.0002 *	0.0073 *	0.0390 *	1.71
BI	0.0000 *	<0.0001 *	0.0945	0.2224	3.62
BBS	<0.0001 *	<0.0001 *	0.0923	1.0000	2.19
TCT	0.3224	0.0001 *	0.0014 *	0.8337	1.64

G1—patients without aphasia; G2—patients with aphasia; BI—Barthel Index; SAS—Sitting Assessment Scale; BBS—Berg Balance Scale; TCT—Trunk Control Test; T1—initial assessment; T2—final assessment; * *p* < 0.05.

**Table 5 jcm-11-03478-t005:** Up & Go and Timed Walk Test—qualitative analyses.

		T1 G1 Group		T1 G2 Group		T2 G1 Group		T2 G2 Group		Fisher’s Exact Test	Cohen’s d
		*n*	%	*n*	%	*n*	%	*n*	%	*p*	
TUG [s]	Not done	37	68.52	49	79.03	13	24.07	22	35.48	<0.0001 *	2.04
	With walking frame	7	12.96	3	4.84	6	11.11	3	4.84		
	With walking stick	1	1.85	0	0.00	7	12.96	3	4.84		
	Independent walking	9	16.67	10	16.13	28	51.85	34	54.84		
TWT [s]	Not done	36	66.67	49	79.03	12	22.22	22	35.48	<0.0001 *	2.20
	With walking frame	8	14.81	3	4.84	7	12.96	3	4.84		
	With walking stick	1	1.85	0	0.00	7	12.96	3	4.84		
	Independent walking	9	16.67	10	16.13	28	51.85	34	54.84		

TUG—Test Up & Go; TWT—Timed Walk Test; G1—patients without aphasia; G2—patients with aphasia; T1—initial assessment; T2—final assessment; * *p* < 0.05.

**Table 6 jcm-11-03478-t006:** Regression analysis examining the effect of the presence of aphasia on BI scores.

BI (T1)	Co. B	±95% CI	*p*	R^2^ Adjusted	SE	Effect Size (f2)
aphasia _1	−15.36	−23.94–6.78	0.0006 *	0.10	23.28	0.11
BI (T2)	Co. B	±95% CI	*p*	R^2^	SE	Effect size (f2)
aphasia _2	−8.36	−20.29–3.56	0.1675	0.02	32.42	0.02

BI—Barthel Index; T1—initial assessment; T2—final assessment; Co. B—slope coefficient; CI—confidence interval; SE—standard error; f2- Cohen’s f-square test; * *p* < 0.05.

**Table 7 jcm-11-03478-t007:** Multivariate regression analysis examining the effects of socio-geographic factors on BI scores at T1 and T2.

	BI in T1			BI in T2		
	Co. B	±95% CI	*p*	Co. B	±95% CI	*p*
Gender	2.31	−6.30–10.92	0.5959	12.12	0.27–23.96	0.0451 *
Age	−0.81	−1.24–0.38	0.0003 *	−0.85	−1.44–0.27	0.0047 *
Education	−0.54	−9.26–8.19	0.9030	0.58	−11.41–12.56	0.9242
Marital status	−9.44	−18.34–0.53	0.0380 *	−3.38	−15.55–8.79	0.5836
Type of stroke	−5.04	−17.40–7.32	0.4205	9.54	−7.43–26.51	0.2676
Time since stroke	−0.33	−0.93–0.27	0.2744	−0.68	−1.49–0.13	0.0987
Length of stay	-	-	-	−0.13	−0.31–0.04	0.1401
Aphasia	−17.03	−25.51–8.56	0.0001 *	−6.99	−18.49–5.41	0.2308
R^2^ adjusted	0.18			0.15		
SE	22.10			30.01		
Effect size (f2)	0.22			0.17		

BI—Barthel Index; T1—initial assessment; T2—final assessment; Co. B—slope coefficient; CI—confidence interval; SE—standard error; * *p* < 0.05.

**Table 8 jcm-11-03478-t008:** Multivariate regression analysis exploring the effects of the studied parameters on BI scores at T1 and T2.

	BI in T1			BI in T2		
	Co. B	±95% CI	*p*	Co. B	±95% CI	*p*
SAS head	−4.07	−11.12–2.98	0.2546	3.39	−8.88–15.65	0.5854
SAS trunk	4.25	−1.89–10.40	0.1727	−7.74	−15.19–0.28	0.0420
SAS arm control	1.01	−4.65–6.68	0.7238	−0.88	−8.46–6.70	0.8186
SAS hand function	−1.85	−7.21–3.52	0.4970	0.21	−6.67–7.09	0.9520
BBS	0.97	0.63–1.32	<0.0001 *	1.57	1.24–1.90	<0.0001 *
TCT	0.02	−0.19–0.24	0.8248	0.43	0.13–0.72	0.0050 *
Aphasia	−5.36	−12.04–1.33	0.1154	−2.92	−8.85–3.01	0.3314
R^2^ adjusted	0.56			0.78		
SE	16.29			15.37		
Effect size (f2)	1.27			3.55		

BI—the Barthel Index; SAS—Sitting Assessment Scale; BBS—Berg Balance Scale; TCT—Trunk Control Test; T1—initial assessment; T2—final assessment; Co. B—slope coefficient; CI—confidence interval; SE—standard error; * *p* < 0.05.

## Data Availability

The data presented in this study are available on request from the corresponding author.

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
