# Peer review of "A Comparative Analysis of Functional Status and Mobility in Stroke Patients with and without Aphasia"

_jcm, 2022, doi:10.3390/jcm11123478_

Round 1

Reviewer 1 Report

I have the following comments to the auhtors' attention:
-The title should be more clear. The reader expects to find data about the functional outcome of the language domain.

Abstract: it is unclear whether the results reported in the abstract are significant or not. Sentences like:
 The group of post-stroke patients with aphasia had a significantly longer elapsed time since their strokes, greater (is that significant?) length of stay in the ward, and worse (is that significant?)  SAS and TCT scores at T2, compared to patients without aphasia., should be clerarly stated.

The conclusion is odd "Aphasia is not an impediment to effective rehabilitation, or a reason to exclude such patients from future studies."

In general, the reader does not weel understand which type of aphasia patients are included, those with production deficit, comprehension deficits, global aphasics. This is a relevant issue in order to understand the results.

There are no study predictions.

In the Methods section among the inclusion criteria, authors report ", patients who were able to understand speech and commands well (patients with no speech comprehension problem"

and "the occurrence of aphasia but not expressive aphasia (Broca’s aphasia)"

these two kind of aphasics are exacly those who can present difficoulties in following a rehab program. It is unclear to me why they were excluded, as they can represent a very relevant picture of aphasia effects on the functional outcome

some lines below it is reported that "G2: 62 patients with expressive aphasia (Broca’s aphasia; no speech comprehension problems). "
thus it is unclear whether Broca's aphasia was or not an exclusion criteria.

In general authors should present some data on patients' language evaluation

Author Response

A comparative analysis of functional outcome in stroke patients with and without aphasia

ID jcm-1727531

We would like to thank the Reviewer for his time and commitment to evaluating our submission. Thank you for pointing to us valuable comments. We are convinced that the changes we have made have improved the article adequately. Any corrections in the manuscript are marked with the "Track changes" function.

Reviewer’s general comment: I have the following comments to the auhtors' attention:
-The title should be more clear. The reader expects to find data about the functional outcome of the language domain.

Author’s response: Thank you for your comment. A lot of researchers agree that aphasia is a serious consequence of stroke, however they report contradictory data regarding the functional status and recovery of patients with and without aphasia after a stroke. The role of aphasia in predicting rehabilitation outcomes after a stroke has not been clearly defined. Therefore we want to inspect the functional status and its improvement in a group of post-stroke patients with and without aphasia, undergoing systematic rehabilitation. This study doesn’t  assess the outcome of language therapy.

The title was changed to: A comparative analysis of functional status and mobility in stroke patients with and without aphasia.

Reviewer’s comment: Abstract: it is unclear whether the results reported in the abstract are significant or not. Sentences like: The group of post-stroke patients with aphasia had a significantly longer elapsed time since their strokes, greater (is that significant?) length of stay in the ward, and worse (is that significant?)  SAS and TCT scores at T2, compared to patients without aphasia., should be clerarly stated.

Author’s response: Thank you for your comments. We agree that the results should be presented more clearly. It was corrected.

Reviewer’s comment: The conclusion is odd "Aphasia is not an impediment to effective rehabilitation, or a reason to exclude such patients from future studies."

Author’s response: We agree that this conclusion is not clearly spelled out. Corrected on: Aphasia should not be an absolute factor which excluding stroke patients from research studies on their functional status.

Reviewer’s comment: There are no study predictions.

Author’s response: Thank you for pointing this. This has been corrected.

Reviewer’s comment: In general, the reader does not weel understand which type of aphasia patients are included, those with production deficit, comprehension deficits, global aphasics. This is a relevant issue in order to understand the results.

In the Methods section among the inclusion criteria, authors report ", patients who were able to understand speech and commands well (patients with no speech comprehension problem" and "the occurrence of aphasia but not expressive aphasia (Broca’s aphasia)" these two kind of aphasics are exacly those who can present difficoulties in following a rehab program. It is unclear to me why they were excluded, as they can represent a very relevant picture of aphasia effects on the functional outcome some lines below it is reported that "G2: 62 patients with expressive aphasia (Broca’s aphasia; no speech comprehension problems)"
thus it is unclear whether Broca's aphasia was or not an exclusion criteria.

Author’s response: We agree with the Reviewer that it is not clear described. It was corrected.

Reviewer’s comment: In general authors should present some data on patients' language evaluation.

Author’s response: The assess the patients' language evaluation was not the aim of this study. The aim of this study was therefore to assess functional status and its improvement in a group of post-stroke patients with and without aphasia, undergoing systematic rehabilitation, and to identify factors affecting the functional status of post-stroke patients at the time of admission and at discharge.

Reviewer 2 Report

It is considered to have clinical significance as a study analyzing stroke patients according to the presence of absence of aphasia. Some revisions are needed.

1. The study period is missing. There is no information on whether all inpatients for a certain period of time were enrolled, or whether there were any patients who dropped out.

2. If not all patients were recruited during the period, the two groups should be matched for recruitment; gender, age, type of stroke, time since stroke, severity...

3. In Table 1, the number of patients by gender does not match. The sum of G1 and G2 and the total number of N are different.

4. In Table 7, did you analyze the time since stroke and length of stay as one?

Author Response

A comparative analysis of functional outcome in stroke patients with and without aphasia

ID jcm-1727531

We would like to thank the Reviewer for his time and commitment to evaluating our submission. Thank you for pointing to us valuable comments. We are convinced that the changes we have made have improved the article adequately. Any corrections in the manuscript are marked with the "Track changes" function.

Reviewer’s general comment: It is considered to have clinical significance as a study analyzing stroke patients according to the presence of absence of aphasia. Some revisions are needed.

The study period is missing. There is no information on whether all inpatients for a certain period of time were enrolled, or whether there were any patients who dropped out.

Author’s response: The study was conducted in the period from January 2019 to February 2022. The  patients after stroke consecutively admitted to rehabilitation ward who met the inclusion criteria were recruited.

This information were completed in article.

Reviewer’s comment: If not all patients were recruited during the period, the two groups should be matched for recruitment; gender, age, type of stroke, time since stroke, severity.

Author’s response: To avoid a small group size, all patients admitted to the ward who met the inclusion criteria were enrolled in the study. The authors realize that this is not a methodologically perfect solution, but it reflects the situation in rehabilitation departments. Thus, the research results may be useful not only from a scientific but also practical point of view.

Reviewer’s comment: In Table 1, the number of patients by gender does not match. The sum of G1 and G2 and the total number of N are different.

Author’s response: Thank you for the remark.  Of course you have right. It was corrected.

Reviewer’s comment: In Table 7, did you analyze the time since stroke and length of stay as one?

Author’s response:  Thank you for the remark.  No, we did not analyze the time since stroke and length of stay as one. It is mistake. It crepted while copying the table into the template.

In part of table: BI in T1 we shown only the time since stroke. In part: BI in T2 we shown in oryginal version only length of stay. In corrected table we added the result of time since stroke.

We also corrected minor errors or deficiencies in the formatting of the remaining tables.

Round 2

Reviewer 1 Report

Authors addressed all the points I rasied. The ms. imporved in clearity.